# The Self and the Other: A Further Reflection on Buddhist–Christian Dialogue

Shiying Zhang

College of Philosophy, Nankai University, Tianjin 300350, China; zhangsy73@163.com

**Abstract:** The dialogue between and comparative research into Christianity and Buddhism theoretically involve the issues of self and other. Faced with the cultural reality of religious diversity, theologies of religions provide four modes of dialogue through which Christianity can interface with religious others. The exploration of the infinite and transcendent traits of otherness in contemporary phenomenological philosophy, as well as the emphasis on differences in postmodern philosophy, contributes to maintaining a clear awareness of otherness and self-identity in the Buddhist–Christian dialogue. Following the dialogical path in comparative theology, which leads one out of oneself, into the other, and back into oneself, in experimental Buddhist-Christian dialogue activities, both Christianity and Buddhism figure as the self and the other. If they openly accept each other's otherness and heterogeneity, view each other as mirrors, and criticize and reflect on themselves, then creative insights into themselves will ultimately be generated. Their selves will be rediscovered, and their understanding and expression will be updated. Reflecting on the Buddhist–Christian dialogue from four aspects, namely, ultimate realism, cosmology, ethics, and religious ideals, can eliminate some misunderstandings and deepen both parties' understandings of themselves and others.

**Keywords:** self; other; Buddhist–Christian dialogue; comparative theology; net of Indra

## 1. Introduction

The practical activities and historical development of religious dialogue tell us that the relationship between the self and the other is an enduring and central issue that cannot be avoided in the practice of religious dialogue. Both parties in a dialogue regard each other as the self and other, and the way in which this relationship is managed significantly affects the effectiveness of the dialogue. Within dialogical activities, both parties maintain a clear self-consciousness, adhering to their own core beliefs, doctrines, rituals, systems, and practices, particularly those non-negotiable concepts that uniquely define their own traditions. They remain loyal to their traditions, preserving their independence, while also maintaining an open attitude towards others, respecting their differences, appreciating their otherness, and engaging with others in a spirit of tolerance and love. Furthermore, regarding the understanding of one's own tradition as an indispensable starting point for dialogue, when facing the other, our immersion in our own traditions, cultures, concepts, and ways of thinking constitutes the predetermined framework through which we perceive religious others. We can only approach the other's traditions and thought world or seek to understand and interpret the other from our own starting point, which is referred to as "prejudices" (*Vorurteile*) by Hans-Georg Gadamer. Lastly, after entering the other's tradition and thought world, instead of translating the other through one's own original textual language, one should strive to understand the other's peculiar thoughts and traits as much as possible according to the other's texts and context. One should also use the other as a mirror to reflect upon oneself, to discover one's relative strengths and weaknesses, and to employ new theoretical perspectives to enhance self-understanding. As a result, a transcendence of the other and a return to the oneself will be achieved.

## 2. Self and Other in the Theology of Religions

The major world religions have long histories and intricate doctrines, offering profound spiritual insights, possessing systematic understandings of their own beliefs, and forming relatively clear self-consciousnesses. However, in the context of interreligious dialogue, the issue of others remains in an experimental phase, with the awareness of others also being in a formative stage. Based on the principles of religious pluralism, Christian theologians have developed various models for interreligious relations, providing theoretical guidance for the practice of dialogue among the world's major religions. Alan Race categorizes the relationships between Christianity and other religions into exclusivism, inclusivism, and pluralism (Race 1983). Paul F. Knitter, in the work "*Introducing Theologies of Religions*", proposes four models: replacement, both holistic and partial replacement, corresponding to exclusivism; fulfillment, akin to inclusivism; mutuality, being divided into philosophical–theological, religious–mystical, and ethical–practical approaches, corresponding to pluralism; acceptance, being divided into post-liberal theology, various salvations; and comparative theology, belonging to postmodern models of interreligious relations (Knitter 2002, pp. 240–43).

Exclusivism completely excludes the importance of the other in terms of truth and salvation, considering itself as unique, while thinking that other religions lack truth and must be replaced. Adherents of other religions must abandon their beliefs and convert to Christianity in order to be saved. Christian evangelicals, fundamentalists, and theologian Karl Barth represent exclusivism. Setting God's revelation in opposition to religion and identifying Christianity with other religions in a religious sense, Barth asserts that Christianity does not possess truth in itself, but only in its inclusion of God's word and revelation by Jesus Christ. Thus, Christianity, in terms of its religious significance, is false. Exclusivists regard other religions as objectified cognitive objects as opposed to the self, rather than as subjects equal to the self in an I–you relationship. The essence of their dialogue essentially adheres to strategies of assimilating others and not respecting or acknowledging the alterity of others, thus making substantial dialogue fundamentally impossible.

Inclusivism treats others with an open and inclusive attitude, overcoming the exclusiveness and particularism of Christ-centrism, through the omnipresence of divine revelation and the Holy Spirit. On the basis of balancing the universality and particularity of divine revelation and grace, it actively engages in dialogue with other religions and discovers hidden self-existence in others. Following the Second Vatican Council, the Catholic Church has regarded interfaith dialogue as a service second only to missionary work. The theologian Karl Rahner introduced the concept of the "anonymous Christian", and said, "Christianity should not simply face members outside the Christian faith as non-Christians, but rather as individuals who can and must be regarded as anonymous Christians in some sense" (Rahner 1966, p. 131). Inclusivists, from their own perspective, acknowledge the relative truth of other faiths or the presence of God in parts of other religions, and recognize the legitimacy of other religions as paths to salvation. They consider adherents of other religions who follow the guidance of conscience and practice good deeds as analogous to Christians, and thus integrate other religions into their own system of values. Essentially, inclusivism regards religious others as another self; adherents of other religions do not possess a truly independent otherness, but rather become a presentation of the self in the guise of the other. In contrast to the absence of the other in exclusivist approaches, in inclusivism, the other exists only nominally; both are forms of religious monism.

The idea of the pluralism of religion is based on the fact that religion is a common spiritual and cultural reality for humanity. Its proponents seek to identify shared points of metaphysical pursuit, mystical experience, and ethical practice among various world religions in order to construct a common platform for interreligious dialogue. Whether it is John Hick's shift from a God-centered to a reality-centered human religious life, Raimon Panikkar's proposition of the basic religious fact of "cosmos-God-man", or Paul Knitter's emphasis on the suffering Earth and the poor, all acknowledge the right of the existence of other religions, and unlike Christianity, they recognize that these are independent paths

and spaces of salvation or redemption. Pluralism in religion, while upholding its own core understandings, maintains openness to others, respects the differences between religions, and rejects the domestication or homogenization of others. Hick, in inheriting the core beliefs of Christianity rather than its historical dogmas, advocates that other religions are human responses to an ultimate divine reality in different cultural contexts. As such, they are seen as worthy of exploration and learning in order to facilitate one's awakening or redemption in life (Hick 1983, p. 121). Panikkar, based on his intuitive insight of "cosmos-God-man", affirms the cosmos, God, and humanity as the otherness of existence, wisdom, and bliss, preserving their differences in an organic and harmonious manner within the *Brahman* itself. Based on the nondualism of *Advaita Vedanta*, Panikkar emphasizes that religious pluralism is neither religious monism nor religious multiplicity but sees the integration of other religious cultures based on the harmony of "cosmos-God-man", allowing differences and communication to maintain a tensional relationship while achieving a harmonious unity of religious cultures. Knitter, working based on the actual issues of suffering and poverty on Earth, calls upon all major religious traditions to take on their global responsibilities, collectively confront the various existential predicaments of humanity, and provide beneficial suggestions or answers from the traditional resources of their respective religions.

In the late 20th century, influenced by postmodernist thought, a mode of acceptance of the relationships between Christianity and other religions was developed by post-liberal theologians represented by Lindbeck and Mark Heim. Lindbeck upheld the slogan "Long live difference", believing that religions have different experiences, no common ground, and are incommensurable. There is an insurmountable cultural–linguistic gap between religions, which cannot be translated. Each religion provides a "comprehensive framework, universal perspective", and a holistic worldview. Each religion cannot be measured by other religions. Heim, on the other hand, believes that religious diversity is based on the diversity of God himself, the triune God as a differentiated community, and that all beings derive existence and life from the differences in relationality. It is precisely these differences that constitute the necessity of dialogue, and many religions have many ways of salvation. Accepting the diversity of others does not contradict manifesting one's own understanding, just as being loyal to one's tradition and remaining open to others are not contradictory. Postliberal theology metaphorically referred to the differences between religions as "fences" and believed that "good fences make good neighbors". Engaging in dialogue across the fence, rather than based on shared mystical experiences and realities, is vital. True differences aid genuine dialogue, and multiple salvations contribute to better dialogue. In this context, both parties in the dialogue maintain a clear self-consciousness while humbly acknowledging that they cannot fully grasp the essential nature of the other.

Represented by comparative theologians Francis X. Clooney, S.J., and James Fredericks, there is a call for religious theologians to suspend the above-mentioned effort at theological construction aimed at facilitating dialogue. When there is limited knowledge or a shallow understanding of other religions, using theology as a microscope to understand and examine other religions appears in some sense to be an attempt to domesticate these religions (Fredericks 1999, pp. 139, 169). Religious theology should result from dialogue and comparison between religions, rather than being a prerequisite for dialogue. During the suspension of efforts to theologize others' religious beliefs, there should be mutual learning. We should form partnerships with others, share the richness of life and spirituality in other religions, and then transcend them and return to oneself. Using other religions as a lens and material for the reinterpretation, critique, and reconstruction of one's own religious tradition can lead to a deeper understanding of oneself. According to Clooney, comparative theology follows the principle of "faith seeking understanding"; its purpose is to better understand the meaning of Christianity. A better understanding of the self comes from a better understanding of others. The realities of other religions, and the new awareness of Christians about them, demand and challenge Christians to develop new ways of understanding Christianity. By restricting research topics to specific texts, spe-

cific rituals, core beliefs, specific theologians, specific backgrounds, and historical periods, through ongoing reflection and comparison, "Comparative theology is the acquisition of a new literacy, in which one's theology is enriched and complicated by reception of the vocabulary, methods, choices of the other tradition, and by one's assimilation of these into one's home tradition. While the elements of one's own tradition may individually remain stable, because located in new contexts, they have 'different meanings'".[1] In a diversified context, comparative theology takes the existence of other religions as a premise, ventures into other religions with a loving, apologetic, and open attitude towards the truth, accepts the challenges of interreligious relations, uses the other as a context, and realizes its own theological understanding and reconstruction.

### 3. Philosophical Understanding of Other

The modern Western cultural construction formed by the Enlightenment places the self and reason at the center of its conceptualization, understanding the self, nature, society, and even the transcendent God in a subjective manner. Despite the tolerance that Enlightenment thinkers hold towards the other, the other is also objectified through the intentional activities of the knowing subject and is thus grasped by thought and internalized as the content of conscious experience. In Kant's epistemology, there is the thing-in-itself (*Ding an sich*) which will never appear in phenomena; it forever remains an unknowable other to the self, ensuring the transcendence of the thing-in-itself (matter or God). Hegel was extremely dissatisfied with Kant's limitation of the rational cognitive ability of humans. In his work, *The Phenomenology of Spirit*, he reveals the dialectical movement of the absolute spirit in the development of consciousness in order to demonstrate the absolute other known to the self and, at the same time, known through the cognitive activities of humans understanding their own selves. The other, possessing transcendence and infinity, gains immanence in concrete reality, pervading human thought, nature, society, and history. Hegel's view of the true infinite, and the view that God cannot exist apart from the world, undoubtedly weakens the absolute transcendence of God in religious belief.

Husserl's concept of transcendence, consistent with Kant's understanding, refers to substance and God. In Husserl's view, the transcendence of substance and God is different. When transcendence is used to describe material things, it refers to the material things being adumbrated and relative; when used to refer to God, the meaning is different. God cannot be given in an adumbrated manner. It is not relative but exists as an absolute divine spirituality and its absoluteness is determined by itself. God shares absoluteness with consciousness and can exist apart from the material world. Hower, God is fundamentally different and cannot be understood by using categories pertaining to the material world and the realm of consciousness. The Hegelian "worldly God" is impossible. The transcendence of God stands opposed to the transcendence of substance; how does God manifest in consciousness? Husserl says that "The 'divine' existence external to the world is such that this existence is as obviously transcendent not only to the world, but also to the 'absolute' consciousness. Hence, it would be such an 'absolute', entirely different from consciousness as absolute, as it would be such a transcendent, entirely different from the world" (Husserl 1983, p. 134). The transcendence of God cannot be understood through the transcendence of the world, and the absoluteness of God cannot be understood through the absoluteness of consciousness; God belongs to a different, divine realm apart from the world and consciousness, manifesting in consciousness according to its own mode of giving.

French phenomenologist Emmanuel Levinas discusses transcendence and infinity from the perspective of ethical relationships. He believes that social relationships between people take precedence over relationships with material objects; individuals do not engage with natural objects in isolation. The transcendent other does not appear as an objective entity; it completely overwhelms thought or intentionality, transcending the scale of consciousness akin to a blinding light affecting the eyes. The experience of transcendence is similarly not an objective experience. "Experience implies a relationship with the absolute other, which is a relationship with something that always overwhelms thought; a relation-

ship with infinity that fully realizes experience" ([Levinas 1969](), p. 25). The infinite as the other is *kath auto*: it exists in itself and relies on itself. The manifestation of the other is similar to epiphany, where transcendence and infinity become related to the self. However, the idea of infinity is placed within self, the self is passive, and the self's concept of infinity does not equate to infinity itself. The concept of infinity always transcends all the concepts formed by the self about it.

Postmodern philosophy criticizes the modern Western pursuit of universal truth and universal values that obliterate the differences between individuals, nations, cultures, and civilizations. It is not advisable to overly rely on the capacity of human reason. Postmodernism argues that difference is a part of life, and that universal truth is dangerous. They propose the slogan "Long live difference" (*Vive la difference*). Respecting differences and allowing them to dominate can prevent different entities from losing their distinctiveness in the process of mutual connection and unification. Allowing difference to govern the unity of thought undoubtedly represents a deep understanding of respect for the other and the transcendence within the other, especially as an absolute transcendent being. Postmodern philosopher Jacques Derrida has said concerning infinity, "Unless his alterity cannot be reduced, in other words, the infinite cannot be reduced, otherwise, other will not become the other; as infinite, other is not merely infinite" ([Derrida 1978](), p. 104). Transcendence determines the otherness of the other, and also sustains the existence of the fundamental differences between things.

### 4. Buddhism and Christianity as Mutual Others

In the history of world religions, Christianity and Buddhism have traditions spanning over two thousand years. Both possess extensive and profound doctrinal systems, which are rich in theological and philosophical thought. As two heterogeneous religious and cultural traditions, representing the East and the West, respectively, their encounter is one of the most intriguing intellectual efforts in this field. As David Tracy stated, "The Buddhist-Christian dialogue has proved to be one of the most puzzling and fruitul attempts in our period. It is an exceptionally fruitful dialogue in sofar as the reality of the other as other is acknowleged as at the heart of all true dialogue." ([Tracy 1990](), p. 68). The historical encounter between Christianity and Buddhism demonstrates that Christian missionaries or theologians predominantly took the initiative in engaging with and understanding Buddhism, this unfamiliar "other".

After the Reformation in the 16th century, Jesuit missionaries passed through India, Sri Lanka, Southeast Asia, and eventually reached Japan, China, and Tibet, initiating encounters between Christianity and the three major traditions of Buddhism. Taking the Jesuits who entered China during the late Ming and early Qing dynasties as an example, in the process of formulating Matteo Ricci's rules, which moved from a practice of attaching Buddhism to preaching to one of "forsaking Buddhism to accommodate Confucianism, supplement Confucianism, and even surpass Confucianism", the "debate between heaven and Buddha" between Jesuits and Buddhist monks reflected the confrontation and conflict between the core doctrines of the two religions. This confrontation included differences between the cosmological perspectives of God's creation and the Buddhist doctrine of interdependent arising, the respective ultimate realities of God and emptiness, perspectives on the compassionate unity and the divine nature of life, differences between the eschatology and the reincarnation (*samsara*) in historical perspectives, and the ethical differences between God's judgment and karmic consequences (*karma*). Based on an exclusionary perspective, the Jesuits dismissed Buddhism as superstitious idol worship, a situation that persisted well into the 19th century. Max Müller, the founder of religious studies, once said that Buddhism was the farthest from truth among religions. In 1876, as part of Müller's leadership in compiling the *Sacred Books of the East*, a large number of Sanskrit and Pali Buddhist scriptures were translated into English. Therefore, Indian Buddhism, Theravada Buddhism, Tibetan Buddhism, and their thoughts were gradually brought into the academia of Western oriental and Asian studies. In the 19th century, with the rise of European Sinology, a large



number of missionaries or professional Sinologists returned to Europe to hold academic positions at renowned universities, such as James Legge, William Edward Soothill, and Richard Wilhelm, who played significant roles in the European translation and study of Buddhist literature. Following the publication of the *Harvard Oriental Series* by American scholar C.R. Lanmann in 1891, influenced by the Japanese Buddhist scholar D.T. Suzuki, American Buddhist studies initially focused on the study of Chinese Buddhism, bringing forth a large number of Buddhist scholars represented by Heinrich Dumoulin, making the United States the center of Western Buddhist studies today. To this day, the academic accumulation of Western Indology, Sinology, Tibetology, Sanskrit and Pali literature regarding Buddhist scripture, history, and philosophy has provided a solid foundation for establishing a deeper and more objective understanding of the cultural spirit and essence of Buddhism as an Eastern other in the Western world.

Inspired by Western Buddhist academic research, the Buddhist cultural sphere has responded based on the comparison between Eastern and Western cultures, presenting its own perspectives on Western culture or Christian civilization. The Kyoto School, represented by figures such as Nishida Kitaro, Tanabe Hajime, Nishitani Keiji, and Abe Masao, responded to the challenges faced by humanity in the pluralistic context of religions, basing their religious philosophy on the communication between Mahayana Buddhism, Zen Buddhism, and classical German philosophy, and proposing the ultimate reality of "absolute nothingness" or "emptiness". The religious philosophy of the Kyoto School involves comparison and dialogue between Buddhist and Christian theology. Tanabe Hajime proposed the concept of God as absolute nothingness, and Abe Masao offered a non-dualistic unity as a basis for dialogue in the context of religious pluralism. The academic dialogue between the scholars of the Kyoto School and prominent Western theologians such as John Hick, Hans Waldenfels, John B. Cobb, and Paul Tillich has advanced the Buddhist–Christian dialogue from mere mutual understanding to mutual learning and transformation. In 1980, the University of Hawaii published the journal, *Buddhist–Christian Studies*, to provide a solid academic platform for comparative research and dialogue between Christianity and Buddhism in Western academia.

When influential Chinese Buddhist monks like Master Xuanhua (宣化上人), Venerable Sheng Yen (星云法师), Thai Buddhist Buddhadasa Bhikkhu, and Vietnamese Zen Master Thích Nhất Hạnh began to preach Buddhism in Europe and America, many Westerners and Asian immigrants became more acquainted with Buddhist ideologies, systems, and life. Influenced by John P. Keenan's advocacy of Mahayana theology, Chinese scholar Lai Pan-Chiu regards Mahayana theology as a type of comparative theology, asserting that it does not treat Mahayana Buddhism merely as a resource, a subject of critique, or a debating opponent. He stated, "The task of Mahayana theology does not necessarily require the use of Mahayana Buddhist language to construct a new Christian theology, but can also reflect, criticize, and reconstruct Christian traditions in accordance with the spirit of Mahayana." (Lai 2011, p. 25) Lai Pan-Chiu elucidates the conceptualization of Mahayana theology from the perspectives of Chinese Buddhist theories, expedient means, Pure Land practices, Huayan Buddhist studies, and the liberation of sentient beings. He rethinks some core doctrines of Christianity and reflects on the methodology of comparative theology.

Under the influence of Western theologies of religions and the comparative philosophies of the East and the West, an increasing number of Buddhist scholars, such as Kristen Beise Kiblinger, have explored the idea of the convergence of the Three Vehicles in the *Lotus Sutra*. They believe that Mahayana Buddhism's skillful development towards Hinayana Buddhism embodies Buddhist ecumenism, as expressed in the *Vimalakirti Sutra*: "If a bodhisattva treads the non-path, it is the path leading to Buddhahood" (Chapter 8, 'The Buddha Way')[2]. From the perspective of the Mahayana Buddhist standpoints of emptiness and non-abiding, all other religions are viewed as certain paths to the truth of the Buddhism. John Markransky argues that the Buddha's skillful means in conveying the truth essentially supports a kind of universal inclusivism, neither simply endorsing pluralism nor upholding exclusivism. The stance of Buddhist inclusivism first addresses

the interpretation of Buddhist classics and the classification of Buddha's teachings. It then illustrates the syncretic coexistence of Indian Buddhism with Brahmanism and Jainism in the Vedic religious context, and finally manifests in the integration of Chinese Buddhism with Confucianism and Taoism during Buddhist Sinicization. Regarding the inclusivity towards other faiths, the argument in the *Diamond Sutra* that "all saints come to know the real through the practice of non-actuality and yet differences arise" (Chapter 7, 'No Attainment and No Teaching') affirms the rationale of religious others in reaching truth and liberation. "The so-called Dharma is no Dharma; that is why it is called Dharma" (Chapter 8, 'Born in accordance with the Dharma')[3] signifies the removal of attachment to the Dharma itself through the wisdom of emptiness and thus maintains a perpetual openness to religious others.

## 5. Some Reflections

Buddhism and Christianity are religious others to each other. By virtue of the theoretical and methodological interplay of comparative philosophy, the philosophy of religion, theologies of religions, comparative religion, and comparative theology, and through a multifaceted comparative study and dialogue of doctrines, thoughts, beliefs, rituals, and institutions, both sides gain deeper understandings of each other's spiritual essence, view each other more sympathetically, enter into each other with appreciation and blessing, appreciate each other's unique spiritual insights, and discover marvelous experiences and insights that their own traditions cannot offer, finally deepening their understanding of themselves with gratitude.

Today, both the Christian and Buddhist communities and academia have discarded some initial rudimentary and superficial views from their first encounters. For example, from the perspective of Buddhism, the monotheistic belief in Christianity has not reached the Buddha's discernment of the true nature of the universe and human life. Relying solely on faith as the means of salvation is evidently a demonstration of a lack of wisdom, and still stays at the highest level of heavenly existence among the six realms of reincarnation. From the perspective of Christianity, the Buddha's doctrine of interdependent arising and the absence of a deity appears to lack an understanding of the religious reality of the universe and human life. The self-reliant nature of liberation through Zen meditation (*Dhyāna* and *Samādhi*) evidently ignores human sinfulness and rejects God's grace, making it difficult to say that Buddhism is a religion. Obviously, these understandings mostly come from employing their respective conceptions and linguistic tools without efforts to delve into each other's traditions and contexts. If one adopts a position of mutual respect and learning, striving to grasp the traditions, doctrines, practices, and essence of others, it becomes apparent that the global philosophy of religion provides a common framework for their comparison on the basis of an ultimate metaphysical reality, cosmology, and ethics, while also presenting serious challenges to both. In this process of comparative study and dialogue, understanding the other can also be accompanied by the possibility of misunderstanding.

First, let us take a look at the understanding of the ultimate reality in both religions. Regarding Christianity, the ultimate reality is God, manifested in the Trinity. Such a deity possesses many attributes, such as being divine, immortal, singular, eternal, formless, impassible, omniscient, omnipotent, omnipresent, benevolent, just, and so on. The essence of God surpasses all human cognition, and to human reason God in itself is an unknown mystery. However, God becomes incarnate for us as a human, having a personal image and being closely related to the world and humanity, with involvement in creation, salvation, and efforts to guide people towards perfection. The contemplation by Christian mystics, Pseudo-Dionysius and Meister Eckhart, on the essence of God manifests the infinite and transcendent metaphysical qualities of God, providing a good perspective for understanding the ultimate reality in Buddhism. As a critic of Brahmanism, Buddhism denies the existence of Brahman, but affirms the eternal reality of *Nirvana*, *Dharmakaya*, *sunyata*, and Buddha-nature in an absolute sense. This led John Hick to consider it to be

a different designation of the ultimate universal reality (Hick 2000, p. 77). Buddhism's affirmation of the ultimate reality is not as clear and singular as that found in Christianity, partly due to the non-metaphysical nature of the Buddha's teachings and the rejection of an inherent existence in all realities. However, with the development of Mahayana Buddhist metaphysical philosophy, various ultimate realities have been constructed. Nirvana, as an ideal goal of practice, is considered an ultimate reality akin to heaven, and the bodhisattva path in Mahayana Buddhism tends to emphasize both limited Nirvana (*Saupādisesā ca nibbā-nadhātu*) and unlimited Nirvana (*anupādisesā ca nibbānadhātu*). The *Prajnaparamita* thought prevalent in Mahayana Buddhisim from the 1st to 3rd century used "emptiness" to refer to absolute reality, stating that "All things arise from interdependent arising, I declare that they are empty, just names provisionally established, and I also declare the middle way" (*Madhyamakakarika: Exposition of the Four Noble Truths*)[4]. All phenomena in the world are dependent on causes and conditions, lacking self-nature. The realities perceived by people are based on verbal designations. Recognizing the two aspects of phenomena is in accordance with the middle way. The fact that things are provisionally established does not mean that emptiness is ultimately existent. Any attachment to emptiness and the objective and substantial understanding of emptiness is erroneous and leads to nihilism. Emptiness indicates the reality of suchness (*Tathatā*), but it is not completely equivalent to absolute existence.

The *Tathagatagarbha* thought prevalent from the 3rd to 5th century proposed that the Buddha-nature is the innate basis for the enlightenment of all sentient beings and that the *Tathagatagarbha* is originally endowed with the four virtues, "*nitya-sukha-atma-subha*". This view seemed to indicate a substantialization of Buddha-nature, leading to criticism from Japanese Buddhist scholars such as Hakamaya Noriaki and Matsumoto Shirō, who viewed it as converging with Brahmanism, contrary to the Buddha's teachings of anat-man, and not truly representing Buddhist dogma (Swanson 1993, pp. 115–49). In fact, the Buddha-nature is also based on two kinds of emptinesses (atman and Dharma), express-ing the affirmation of the state of liberation rather than using negative language to express suchness as silence, anatman, emptiness, and transience in the Madhyamika School. There-fore, in the Buddhist context, the appropriate expression for the ultimate reality should be the concept of suchness. It represents the realm of enlightenment and is the truth of the universal and of life.

The concept of "emptiness" is the core idea of Prajnaparamita thought in Mahayana Buddhism. The Kyoto School, inspired by the spirit of Zen emptiness, proposed the idea of "absolute nothingness", transcending the distinction between existence and non-existence, potential and actual, and subject and object. It initiated a dialogue with Western existen-tialism in an attempt to overcome the spiritual crisis brought about by nihilism in the West. Western Christian theology and doctrine have long been shaped by concepts and terms from Greek philosophy such as *ousia, hypostasis, prospon*, etc. Heidegger states that the internal structure of Western metaphysical philosophy is "onto-theo-logie", and that this has determined the way of thinking in Western theology. Re-evaluating Christian theol-ogy by embracing the sunyata, which eradicates all attachments and dichotomies, offers significant edification. By using the concepts and terms of non-substantialism and relation-alism to re-understand and interpret the Christian faith, a form of Christian theology may come to possess more vitality in a non-Western cultural context. By eliminating various personifications, linguistic choices, and conceptual restrictions on God, new insights about God himself might emerge. For instance, Don Cupitt, holding a postmodern position of anti-realism and metaphysics, believes that the Western understanding of God is a cultural product of Western language and that, borrowing the viewpoint of Nagarjuna's emptiness, it is entirely possible to understand God as emptiness (Cupitt 2003, p. 16). The world itself is contingent, and there is no distinction between God and emptiness. Westerners should learn to accept the fact of God's death and the end of the universe. There is no need to fear emptiness; instead, one should dwell in emptiness peacefully and experience the bliss of emptiness.

Furthermore, in terms of cosmology, Christian creation theory explains that all things are created by God, who sustains the order and stability of the world, imparting laws to determine the functioning of things. The theistic tradition of Christianity has always adhered to the essential difference between the Creator and the created world, thereby ensuring the divine transcendence of God. Buddhist cosmology, based on the theory of interdependent arising (*pratītya-samutpāda*) and the doctrine of *anatman* (non-self), uses interdependent arising to explain the causes of all phenomena. Interdependent arising asserts that all phenomena arise due to causes and conditions, depend on interdependent relationships and conditions, and are subject to the law of cause and effect. In the Buddhist theory of interdependent arising, the original "*pratītya-samutpāda-aṅga*" belongs to the arising of karma and feeling. Mahayana Buddhism's theories of "*Tathagata-garbha*" (如来藏), "*dharma-dhatu*" (法界), and "*alayavijnana*" (藏识) are used to address the relationship between noumenon and phenomenal world. According to these theories, everything arises from the coincidence of mental consciousness with external conditions, and all phenomena are impermanent, undergoing the processes of arising, abiding, changing, and ceasing, while the eternal *Tathagata*, *alayavijnana*, *Buddha-dhatu*, and *dharma-dhatu* are inherent in all phenomena. The Buddhist notion of "Oneness of substance and function (体用一如)" shares similarities with modern Christian panentheism (the belief that God is immanent in the world). Panentheism emphasizes the interactivity of God and the world, highlighting the ultimate reality's immanence and downplaying transcendence, with God being present in all things and inseparable from them.

Once again, in terms of ethics and morality, Christianity relies on the doctrine of God's final judgment and the commandments of God. This faith relies on God's grace to absolve sins, teaching people to obey God's will and continuously sanctify themselves, thereby becoming perfectly righteous in the eyes of God and ultimately achieving salvation and enjoying eternal life. Buddhism, based on the doctrines of karma and reincarnation, adheres to numerous precepts, such as earnestly practicing in accordance with teachings, achieving complete awakening, and accumulating merit, in order to ultimately attain Buddhahood. As a religion of love, Christianity requires Christians to imitate Christ and become Christ to their neighbors, sacrificing themselves, serving others, and embodying God's universal love (*Agape*). Mahayana Buddhism particularly emphasizes the boundless compassion of Bodhisattvas; the Buddha himself harbored a compassionate aspiration for all beings to attain the wisdom of liberation. As such, he preached the Dharma and enlightening sentient beings to the following information: "All tathagatas manifest in the world for only one great reason" (*Lotus Sutra*, Skillful Expedient). Mahayana Buddhism posits the equality of all sentient beings, advocating for the arising of great compassion for all beings, with all non-human lives treated kindly. Its understanding of equality and compassion surpasses that of Christianity, extending to encompass the entirety of universal life. However, Buddhism's concept of the equality of all sentient beings and the interconnectedness of all things actually diminishes individual personality and a sense of dignity, which is a fundamental value of modern society and comes from the Christian "image of God (*imago dei*)".

Finally, regarding the theory of religious ideals, the contrast in the notions of Christianity's salvation through reliance on the grace of others and Buddhism's self-attained liberation are evident. However, the doctrine of "*sola gratia*" predominantly asserted by Protestantism stands in stark contrast to the viewpoint advocated by Catholicism, which affirms a collaborative effort of "human cooperation with God for salvation" and acknowledges all efforts made by humans for their own salvation. The combined cultivation of wisdom and insight, leading to self-attained liberation, is universally accepted among the various sects of Buddhism. However, the Pure Land Sect, including its Japanese branch, Jodo Shinshu, advocates "faith and vows", relying on the assistance of the vows of Buddhas and Bodhisattvas to attain liberation, which resembles the mode of salvation followed in Christianity. The Pure Land Sect has opened the gateway to the faith of liberation, yet it differs in its understanding of faith from Christianity. The *Awakening of Faith in Mahayana* (*Mahāyāna śraddhotpada śāstra*) discusses the four correct kinds of faith: the fundamental

Tathatā, the Buddha, the Dharma, and the Sangha. Genuine faith, also known as pure faith, requires insight from wisdom in order to realize the truth. The faith in the Pure Land refers to a firm belief in the existence of the Pure Land, as expounded by Shakyamuni Buddha, the inexhaustible power of Amitabha Buddha's vows, and the absolute guidance of the three saints at the time of death. The Pure Land Sect believes that genuine faith can be divided into true belief (谛信) and sincere belief (仰信). The former refers to the understanding the principles of the Pure Land and the Buddhist teachings, while the latter refers to those who cannot comprehend the meaning of the Buddhist teachings, yet still maintain unwavering belief without any doubt (Zhang 2018, pp. 344–45). Buddhism does not set wisdom against faith; instead, wisdom strengthens faith, and faith encompasses understanding, and both are interwoven into the Buddhist life and practice. The spirit of medieval scholasticism, "faith seeking understanding" (*fides quaerens intellectum*), is a perfect echo of Buddhism's emphasis on genuine faith. Although the grace embedded in the Christian faith also involves acknowledging the Savior, Christian faith leans more toward the distinction between the human and divine, with faith being a supernatural faculty that surpasses reason and necessarily indicates the inadequacy of human cognition. However, in the relationship between faith and religious conduct, Buddhism and Christianity share similar views: "Faith is the source of way and mother of merits. It nourishes all good roots" (*The Avatamsaka Sutra*, Chapter 12). This means that all meritorious deeds originate from genuine faith. The leader of the Protestant Reformation, Martin Luther, in *On Good Works*, explicitly outlined the viewpoint that good deeds stem from faith. This became a classic standpoint among reformers regarding the relationship between faith and love, and the merits of good deeds (Luther 1955, pp. 23–24).

## 6. Conclusions

The dialogue between and comparative study of Christianity and Buddhism represent academically challenging enterprises concerning the human intellect. For a substantial dialogue to occur between these two highly heterogeneous religions and cultural systems, careful consideration of the relationship between the self and the other is necessary. Phenomenology's philosophical exploration of the infinite and the transcendent, as well as postmodernism's emphasis on difference, offer excellent theoretical perspectives for understanding the otherness present in both traditions. Theologies of religions provide a solid theoretical and conceptual basis for the dialogue between Buddhism and Christianity in this era of religious pluralism, while religious philosophy offers a common domain and topics for understanding the religious other. Comparative theology's movement from the self, to the other, and back to the self is of significant guiding importance for the Buddhist–Christian dialogue. Both Buddhism and Christianity are seen as others to each other, and through an experimental dialogue they accept the transcendence and infinity within the other, maintaining an attitude of humility, learning, and appreciation. This approach entails relinquishing inclusive and hegemonic positions that seek to assimilate the other's thought processes and theological systems. Carefully engaging with each other's faith, doctrines, theology, rituals, institutions, and religious practices, and making an effort to understand each other's scriptures and languages, is essential for grasping the spiritual characteristics of the other religion. Respecting the otherness of others involves not viewing them as a resource or a tool but rather as a mirror through which one can reflect on oneself, engaging in self-critique and reflection. This can ultimately lead to creative insights being developed into the self, thereby rediscovering and deepening self-consciousness, and renewing self-understanding and expression.

We believe that once Buddhism and Christianity can embrace each other's otherness, a new understanding will surely shed fresh light on them. The achievements of Mahayana theology and the Kyoto School indicate that accepting otherness and renewing self-understanding are interrelated and mutually supportive, contributing to the positive development of the Buddhist–Christian dialogue. With the increasing attention paid by Western Christianity and Buddhist scholars to the classics, ideological connotations, and theoretical

characteristics of Chinese Buddhism, coupled with more involvement of Chinese religious scholars in the theory and practice of the Buddhist–Christian dialogue, global Buddhist–Christian dialogue will undoubtedly become more diverse and enriching. We draw on the metaphor of Indra's Net from the *Avatamsaka Sutra*, where each knot of the net is adorned with countless jewels, each reflecting the image of all other jewels, This reflection of reflections continues infinitely, leading to the development of an unimpeded and harmonious state.[5] In the dialogue and comparative activities between Buddhism and Christianity, each aspect of religious tradition—thoughts, beliefs, doctrines, concepts, rituals, institutions, and paths of cultivation—is like a jewel in Indra's Net, being capable of reflecting the self-image of both parties while continually deepening religious self-consciousness and understanding. Thus, a better mutual understanding can be achieved and a perfect realm of dialogue between Buddhism and Christianity can be reached.

**Funding:** This project was funded by the National Social Science Foundation of China's general project "Translation and Research on Martin Luther and Modern Thoughts" (23BZJ059), and also by Liberal Arts Development Fund of Nankai University "On St. Anselm's Philosophy of Religion" (2022).

**Institutional Review Board Statement:** Not applicable.

**Informed Consent Statement:** Not applicable.

**Data Availability Statement:** No new data were created or analyzed in this study. Data sharing is not applicable to this article.

**Conflicts of Interest:** The author declares no conflict of interest.

## Notes

[1] For more information about comparative theology, please see *What Is Comparative Theology?* Available online: https://www.bc.edu/bc-web/schools/morrissey/departments/theology/areas-of-study/comparative-theology/what-is-comparative-theology.html (accessed on 25 January 2024).

[2] For more detailed descriptions, please see《维摩诘经》第八品，"文殊师利问维摩诘言：'菩萨云何通达佛道？'维摩诘言：'若菩萨行于非道，是为通达佛道。".

[3] For more detailed explainations, please see 《金刚经》第七品，"所以者何。一切贤圣，皆以无为法而有差别"。第八品，"须菩提。一切诸佛。及诸佛阿耨多罗三藐三菩提法。皆从此经出。须菩提。所谓佛法者。即非佛法。".

[4] For more detailed information, please see龙树：《中论颂》第二十四品，"因缘所生法，我即是空，亦名是假名，亦是中道义。未曾有一法，不从因缘生，是故一切法，无不是空者。".

[5] For deep understanding on the net of Indra, plese see 《华严经》十地品、不思议品和舍那品，《华严五教章》卷四、《华严经探玄记》卷一，《华严经随疏演义钞》卷十.

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
