# Peer review of "The Self and the Other: A Further Reflection on Buddhist–Christian Dialogue"

_religions, doi:10.3390/rel15030376_

Round 1

Reviewer 1 Report

Comments and Suggestions for Authors

The theme of Christianity’s discontinuity and continuity with other religions is dealt (though the terms “continuity” and ‘discontinuity” are not employed) via viewpoint of self and other. 

The argument seems to be in support of inter-religious dialogue based on theological pluralism. The author has woven in the general issues and approaches into the discussion; the clarity of thought-development leading to the conclusion needs improvement. It is suggested that the arguments leading toward the conclusion be further strengthened, references improved, and the bases for Christianity’s self-understanding with reference to the multiplicity of Buddhist schools be given proper theological grounding. Probably, limiting the research to mainly a discussion of one theological approach and its encounter with one particular school of Buddhist philosophy might help to keep the focus clear and enrich the content.

Citations needed (some examples)

Line 37, 53, 55, 79, 101, 136-38, 147, 243, 408-410

Leap of thought occurs in Lines 399-400 from Greek influences directly to Heidegger’s. Is reference to Heidegger indispensable here? Needs explanation.

Revise terminology: Line 339, 415ff “dependent arising” or “dependent co-arising”?

Reviewer 2 Report

Comments and Suggestions for Authors

This is an interesting paper that is suitable for publication in religions. A few minor changes will improve it significantly.

It would help to see engagement with books such as Paul Knitter Without Buddha I could not be a Christian or Paul Knitter and Roger Haight’s Jesus and Buddha, or the edited collection Buddhist-Christian Dual Belonging or Buddhism and Christianity in Dialogue (Schmidt-Leukel 2005) or other such volumes.

The section on exclusivism, inclusivism and pluralism also needs refining, as it does not present the typology accurately. See for example Only One Way? (2011), co-authored by Gavin D’Costa, Paul Knitter and Daniel Strange or Gaston, R., 2017, Faith, Hope and Love: Interfaith Engagement as Practical Theology.

On line 229 it is inaccurately stated Christianity is “Western.” Some traditions are, but others are “Eastern,” such as Thomist Christians in India.

Some technical terms are introduced without explanation (lines 455 or 474 for example) and a quote ends on line 178, but its unclear where it starts.

Comments on the Quality of English Language

There are a few errors, such as line 49 "the otherness", omiision of "they are" at the end of line 157, or "rely" instead of "relies" in line 206.

Reviewer 3 Report

Comments and Suggestions for Authors

The paper is an interesting one and, which in many ways, is helpful in relation to Christian-Buddhist dialogue.

The title of the paper, its introduction and general presentation appear to present its content as if it might be written from a particular (Buddhist or Christian) perspective. If that is intended to be the case, it would be better if that could be clearly articulated in the introduction to the article. However, on reading further into the text, the sense of at least this reviewer is that this piece is, at the very least, written by a Christian author and in many ways at least seems to embody and reflect a Christian conceptual starting-point. If that is indeed the case, it would be best if such "positionality" could at the least be acknowledged. Indeed, it would even be possible for it legitimately to be not just acknowledged but also fully "embraced" if that is indeed the authorial angle from which it is intended to make this contribution.

In terms of detailed textual points, I would highlight two:

1.) When speaking of both Christianity and also Buddhist, at line 47, "theological insights" are referred to, whereas Buddhism in at the least an "a-theistic" religion. This detailed point is one of a number of reasons arising from the text as written for why this reviewer also has the overall perception referred to in the above.

2.) In relation to the Christian theological model of "exclusivism" mentioned at line 61, I would question how it is accurate to claim that this model "completely disregards the existence of the other"? Potentially what it does do is to exclude the validity of any salvific claims on the part of the "other" religion (in this case Buddhism), but that is not the same things as saying that the model "completely disregards" Buddhism. A more careful formulation would therefore be welcome at this point.

Comments on the Quality of English Language

There do appear to be within the text one or two places where a more careful proofing of the English language (in particular with regard to use of the definite article) would improve the text.

Round 2

Reviewer 1 Report

Comments and Suggestions for Authors

Grammatical, spelling checks have been adequately made. The article has also made some changes in use of terminology, e.g. interdependent arising. 

The paper makes a comparative and reflective contribution to the ongoing theme of Christian-Buddhist dialogue. The overview is general and notes insufficiencies in current attempts to dialogue. The main argument may need further exploration and development.